# Imitation dynamics on networks with incomplete information

Xiaochen Wang [1,6], Lei Zhou [2,6], Alex McAvoy [3,4] & Aming Li [1,5] ✉

Imitation is an important learning heuristic in animal and human societies. Previous explorations report that the fate of individuals with cooperative strategies is sensitive to the protocol of imitation, leading to a conundrum about how different styles of imitation quantitatively impact the evolution of cooperation. Here, we take a different perspective on the personal and external social information required by imitation. We develop a general model of imitation dynamics with incomplete information in networked systems, which unifies classical update rules including the death-birth and pairwise-comparison rule on complex networks. Under pairwise interactions, we find that collective cooperation is most promoted if individuals neglect personal information. If personal information is considered, cooperators evolve more readily with more external information. Intriguingly, when interactions take place in groups on networks with low degrees of clustering, using more personal and less external information better facilitates cooperation. Our unifying perspective uncovers intuition by examining the rate and range of competition induced by different information situations.

Quantitatively understanding the evolution of collective behaviour in animal and human societies is a fundamental question in modern science[1–3]. Evolutionary game theory provides a prominent mathematical metaphor to quantify behavioural strategies of individuals, related payoffs, and how they change under the influence of natural selection[4–9]. Unlike in unstructured populations where natural selection favours free riders[10–12], network structure serves as a basic mechanism that promotes cooperation[13] by non-random and local interactions[14–18]. The basic intuition dates back to Hamilton[19,20], who argued that the "viscosity" arising from limited (i.e., local) dispersal leads to altruists benefiting from proximity to genetic relatives. This intuition has been profoundly influential in evolutionary theory, and it is partially responsible for our current understanding of how cooperation evolves in networked systems.

When scrutinizing the evolution of collective cooperation on networks, researchers find that one of the key factors that determine the fate of cooperation is the update rule, i.e., the rule that specifies how individuals change their strategies over time[14,17,21]. Indeed, network structures and update rules are two sides of a coin, with the former acting as the substrate and the latter driving the evolution of the entire system. Imitation-based update rules are commonly used rules in previous studies since imitating successful peers via social comparison is an important learning heuristic in both animal and human societies[22,23]. Intriguingly, previous studies have shown that whether cooperation evolves depends sensitively on the protocol of imitation: forgoing one's own strategy and imitating successful neighbours by comparing all neighbours' payoffs (the so-called "death-birth" update rule) makes cooperation evolve if the benefit-to-cost ratio is greater than a positive threshold; comparing the payoff of a random neighbour with one's own and imitating based on this payoff difference (the so-called "pairwise-comparison" update rule) instead makes cooperation always disfavoured by natural selection irrespective of the benefit-to-cost ratio[14,15,24]. Such qualitatively different results induced by distinct mechanisms of imitation raise important questions

[1]Center for Systems and Control, College of Engineering, Peking University, Beijing 100871, China. [2]School of Automation, Beijing Institute of Technology, Beijing 100081, China. [3]School of Data Science and Society, University of North Carolina at Chapel Hill, Chapel Hill, NC 27599, USA. [4]Department of Mathematics, University of North Carolina at Chapel Hill, Chapel Hill, NC 27599, USA. [5]Center for Multi-Agent Research, Institute for Artificial Intelligence, Peking University, Beijing 100871, China. [6]These authors contributed equally: Xiaochen Wang, Lei Zhou. ✉e-mail: amingli@pku.edu.cn

about how the mechanisms of imitation influence the evolution of cooperation. So far, few studies provide clear and satisfactory answers.

To address these questions, we start by examining the information required by different imitation-based update rules. Two kinds of information are considered, personal and external social information. The former refers to an individual's own strategies and payoffs while the latter refers to those of one's neighbours. From this perspective, the aforementioned two update rules (and other classical imitation-based rules as well) can be clearly differentiated: the death-birth update rule requires no personal information but full social information; the pairwise-comparison update rule needs both personal and social information and weights them equally. This suggests that the amount of personal and social information required and the relative weighting of personal to social information may serve as indicators to quantify the impact of imitation-based update rules on the evolution of cooperation. To undertake a thorough investigation, we propose a new class of imitation-based update rules called "imitation with incomplete social information". Under this rule, the amount of personal and social information and the relative importance of personal to social information during strategy updating are all tunable, covering a wide range of information requirements for strategy updating and recovering classical imitation-based update rules as special cases.

Employing this new class of update rules, we first derive analytical conditions for cooperation to prevail over defection in pairwise social dilemmas. These conditions reveal that it is best for the evolution of cooperation if individuals ignore their own information and instead imitate more successful social peers, irrespective of the number of peers (at least two) used for comparison. In group social dilemmas, the same result holds if the degree of clustering in the network is sufficiently high; otherwise, it is better to rely more on personal information and use less social information. This finding arises mainly from the low overlap between individuals' first-order and second-order neighbours, which makes it easier for defectors to exploit cooperators through group interactions when the network is sparse. Finally, we demonstrate that our findings are robust to heterogeneity in network structure as well as to the individualized utilization of social information. Our results thus highlight the degree to which social information affects the evolution of collective cooperation in networked systems.

## Results

### Games and payoffs

We are interested in conflicts of interest arising in groups. Consider a group of size $n$, consisting of individuals of type C ("cooperator" for example) or D ("defector" for example). Suppose that $f_C(n_C)$ and $f_D(n_C)$ are the respective payoffs to types C and D when there are $n_C$ total cooperators in the group. A simple but highly influential model of a social dilemma was proposed by Dawes[25] as possessing two properties: all individuals prefer widespread cooperation to widespread defection ($f_C(n) > f_D(0)$), yet for all $n_C \leqslant n$, there is a temptation to defect ($f_C(n_C) < f_D(n_C - 1)$). Here, we consider two kinds of these social dilemmas: a donation game, which involves pairwise interactions, and a public goods game, which involves interactions in larger groups.

The networked system we consider consists of $N$ individuals arranged on the nodes of a network, whose structure represents the relationships between individuals. At each time step, individuals interact with neighbours and obtain payoffs from these interactions. In the donation game, every individual interacts with each neighbour separately[12,14,18]. Cooperators (C) pay a cost of $c$ to provide a neighbour with a benefit $b$, while defectors (D) pay no costs and provide no benefits. This pairwise "donation game" can be summarised by the payoff matrix[15,26,27]

$$\begin{array}{cc} & \begin{array}{cc} C & \quad D \end{array} \\ \begin{array}{c} C \\ D \end{array} & \begin{pmatrix} b - c & -c \\ b & 0 \end{pmatrix}, \end{array} \qquad (1)$$

where each entry gives the payoff to the row player against the corresponding column player.

Instead of interacting with each neighbour separately, each game could consist of group interactions, wherein every individual organises a multi-player game[28,29] involving all of its neighbours. If an individual has $d$ neighbours, then they participate in $d + 1$ group interactions, with one initiated by the focal individual and $d$ initiated by the neighbours. A cooperator pays a cost $c$ in each game, and the total costs from cooperators are then enhanced by a multiplication factor and divided among all members of the group. When there are $n_C$ ($0 \leqslant n_C \leqslant n$) cooperators in a group of $n$ individuals, the respective payoffs for defectors and cooperators are

$$f_D(n_C) = \frac{r n_C c}{n}, \qquad (2a)$$

$$f_C(n_C) = f_D(n_C) - c, \qquad (2b)$$

where $r$ is the multiplication factor for the public good. When $1 < r < n$, the players in this game are confronted with a social dilemma, wherein the strategy to maximise individual payoffs (namely, defection) deviates from the collectively optimal choice (namely, cooperation).

In either kind of social dilemma, an individual $i$'s payoff, $u_i$, is calculated as the average of their payoffs over all interactions. This payoff is then transformed into fitness, $F_i$, by the mapping $F_i = e^{\delta u_i}$, where $\delta \geqslant 0$ is the intensity of selection[12,15,18]. The selection intensity reflects the contribution of game interactions to the fitness of $i$, which we assume to be weak. The case of neutral drift corresponds to $\delta = 0$, where cooperators and defectors are indistinguishable from the standpoint of reproductive success.

### Imitation dynamics

Imitation-based rules are commonly used in exploring the evolution of cooperation on complex networks[5,14,18,26,30]. Instead of viewing behaviour change as a result of death, birth, and replacement, imitation models have the property that agents remain alive but can periodically copy the behaviour of others. Popular update rules such as "death-birth" (DB) and "imitation" (IM)[7,14,17,18,24,26,31,32], as well as "pairwise-comparison" (PC)[15,21,33,34], all have natural interpretations in terms of strategy revision in a cultural context[30]. However, these update rules (Fig. 1a–c) lie on the extreme ends of a spectrum in that they assume an individual has access to information about either all neighbours (DB and IM, with the distinction being that imitation is not compulsory under IM) or only one neighbour (PC) when making a decision about whether (and whom) to imitate.

After interactions, a single individual $i$ is selected uniformly at random to update its strategy. The set of neighbours of $i$ whose information (including strategies and payoffs at the current time step) is accessible to $i$ is denoted by $\Omega_i$. We note that $j \in \Omega_i$ only if $j$ is a neighbour of $i$, so $|\Omega_i| \leqslant d_i$, where $d_i$ is the degree of $i$. If this inequality is strict, then $i$ has incomplete social information during imitation. Once social information is determined, the relative importance of $i$'s personal information is quantified by $\theta \in [0,1)$. For any $j \in \Omega_i$, the weight associated to $j$ is $(1 - \theta)/|\Omega_i|$, so the total weight associated to all neighbours for comparison is $1 - \theta$. Under the "imitation with incomplete social information" (abbreviated as "IMisi") update rule

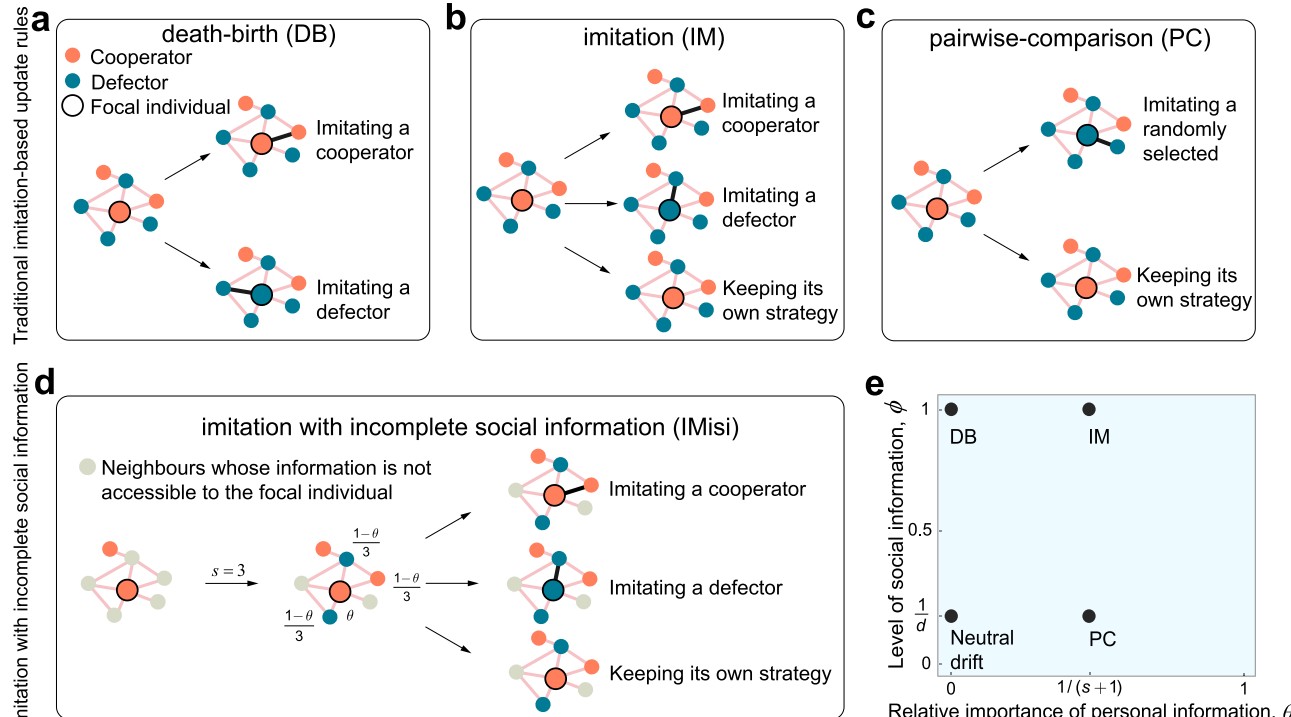

**Fig. 1 | Illustration of imitation dynamics with incomplete information. a** Under the death-birth (DB) update rule, an individual $i$ (denoted as the focal individual marked by a black circle) is randomly selected to update its strategy, and it forgoes its own strategy and imitates its neighbours with a probability proportional to their fitness[15]. The link between $i$ and the neighbour that $i$ imitates is highlighted by a bold black line. The orange and blue filled circles represent cooperators and defectors, respectively. **b** Under the imitation (IM) update rule, an individual $i$ is selected at random to evaluate its strategy. The individual either keeps its current strategy or imitates a neighbour's strategy with a probability proportional to fitness[54,55]. **c** Under the pairwise-comparison (PC) update rule, an individual $i$ is chosen randomly to evaluate its strategy, and a neighbouring individual is chosen at random as a role model[33,34]. Individual $i$ either adopts this neighbour's strategy or retains its own with a probability proportional to their fitness. **d** Under imitation with incomplete social information (IMisi), only $s$ (of $d = 5$) neighbours' information is accessible to the focal individual, and the relative importance of $i$'s personal to external social information is quantified by $\theta$ (namely, the weights for all accessible neighbours are identical and equal to $(1-\theta)/s$). Here, neighbours whose information is not accessible to the focal individual are represented by grey filled circles. Under the IMisi rule, the focal individual could imitate a cooperative or non-cooperative strategy from $s$ neighbours or keep its own strategy (see Eq. (3) for corresponding probabilities). **e** The IMisi rule is a general imitation-based update rule that unifies classical rules including DB, IM, and PC by adjusting the value of $\theta$ and the level of social information $\phi = s/d$: DB ($\phi = 1, \theta = 0$), IM ($\phi = 1, \theta = 1/(d+1)$), neutral drift ($\phi = 1/d, \theta = 0$), and PC ($\phi = 1/d, \theta = 0.5$).

(Fig. 1d), $i$ imitates the strategy of $j \in \Omega_i$ with probability

$$\frac{(1-\theta)F_j}{(1-\theta)\sum_{k \in \Omega_i} F_k + \theta|\Omega_i|F_i}. \tag{3}$$

Otherwise, individual $i$ does not imitate anyone and retains its own strategy. For complete social information ($|\Omega_i| = d_i$), IMisi reduces to the canonical DB ($\theta = 0$) and IM ($\theta = 1/(d_i+1)$) update rules. PC corresponds to $|\Omega_i| = 1$ and $\theta = 1/2$.

For the parameters of interest, this update rule defines an absorbing Markov chain, which eventually ends in a state where all individuals take the same strategy (either all-C or all-D). As a result, we consider the fixation probability of cooperators (resp. defectors), $\rho_C$ (resp. $\rho_D$), which represents the probability that one randomly-placed cooperator (resp. defector) invades and replaces a population of defectors (resp. cooperators). The metric we use to evaluate whether selection favours cooperators over defectors is the value of $\rho_C - \rho_D$. Specifically, cooperators are favoured relative to defectors[11,14,18] if $\rho_C > \rho_D$. Under neutral drift ($\delta = 0$), both $\rho_C$ and $\rho_D$ take the value $1/N$. We note that for the class of imitation dynamics we consider, under weak selection, the condition $\rho_C > \rho_D$ is equivalent to the commonly-used alternative condition $\rho_C > 1/N$, which measures the effects of selection on $\rho_C$ relative to its neutral value[14,18,35].

Here, we are primarily interested in sets $\Omega_i$ that are chosen randomly, subject to the constraint of having fixed size $|\Omega_i| = s$ for some

parameter $s$, which represents the amount of social information. When individuals neglect personal information ($\theta = 0$) and randomly select one neighbour to imitate at each time step ($s = 1$), the evolutionary process is equivalent to neutral drift and is independent of $\delta$. Therefore, we mainly focus on the cases where either $\theta > 0$ or $s > 1$. For the sake of simplifying the expressions we present, we assume that the network is unweighted, undirected, and regular of degree $d$, with no self loops. This assumption is not crucial for deriving results on the IMisi rule; but, in line with previous studies[14,24,36], we find this assumption to be useful to provide intuition for the impact of incomplete information on the evolution of cooperation for the imitation processes we consider. At the conclusion, we briefly consider heterogeneous networks.

**Pairwise social dilemmas**

To investigate the influence of incomplete information on the fate of cooperators, we first consider IMisi for pairwise interactions on regular graphs. For the cases where either $\theta > 0$ or $s > 1$, we show in Methods that weak selection favours cooperators over defectors whenever $b/c > (b/c)^* > 0$, where

$$\left(\frac{b}{c}\right)^* = \frac{2\theta(N-1) + (1-\theta)\frac{1}{s}\frac{s-1}{d-1}d(N-2)}{-2\theta + (1-\theta)\frac{1}{s}\frac{s-1}{d-1}(N-2d)}. \tag{4}$$

To investigate how $\theta$ and $s$ affect $(b/c)^*$, we start from the scenario where individuals ignore their own information during strategy

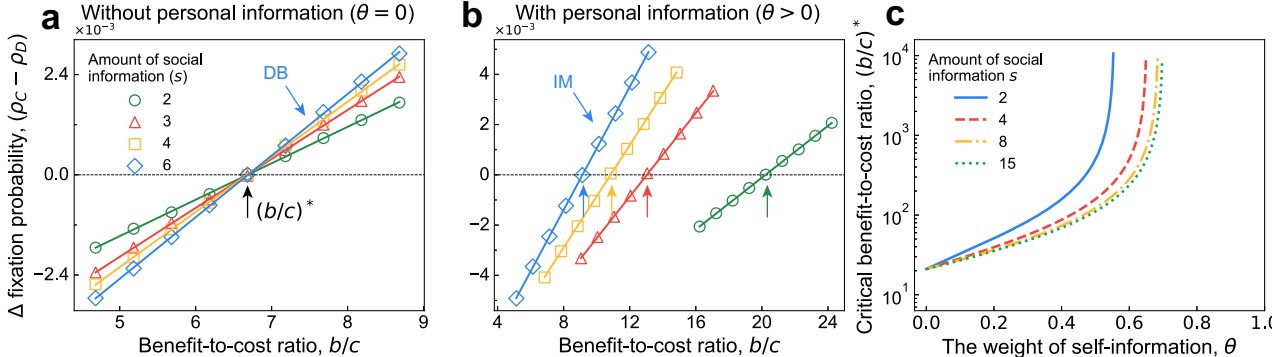

**Fig. 2 | Effects of incomplete information on the fixation of cooperation in pairwise social dilemmas.** Here, we present simulations of the fixation probability difference $\rho_C - \rho_D$ of cooperation and defection as a function of benefit-to-cost ratio, $b/c$. Markers are from numerical simulations and lines from the corresponding linear curve fitting. The vertical arrows point to the values of $(b/c)^*$ derived theoretically under weak selection (Eq. (4)). **a** If individuals ignore their own information ($\theta = 0$), then the $(b/c)^*$ above which cooperation is favoured is the same for different amounts of social information $s > 1$. **b** When individuals treat social and personal information as being equally important ($\theta = 1/(s+1)$), $(b/c)^*$ decreases as $s$ grows. **c** We also illustrate $(b/c)^*$ as a function of the weight of personal information, $\theta$, for different amounts of social information, $s$, according to Eq. (4). All curves converge to the same value of $(b/c)^*$ as $\theta \to 0$, suggesting that cooperation is favoured most when individuals neglect their personal information. Here, we set $N = 100$, $\delta = 0.01$, and $d = 6$ in **a** and **b**, and $d = 15$ in **c**.

updating ($\theta = 0$). In this case, Eq. (4) reduces to $(b/c)^* = d(N-2)/(N-2d)$ for any $1 < s \leqslant d$, and we find that the amount of social information used during imitation has no impact on the fate of cooperators (Fig. 2a). Note that the canonical DB rule[14,24,26,36] is a special case of IMisi with $s = d$ and $\theta = 0$ (Fig. 1e), and for large populations, we obtain a well-known rule[14], namely $\lim_{N \to \infty}(b/c)^* = d$.

When individuals treat all information the same (including both personal and social information), meaning $\theta = 1/(s+1)$, we obtain

$$\left(\frac{b}{c}\right)^* = \frac{(ds+d-2)N - 2ds + 2}{(s-1)N - 2ds + 2}. \quad (5)$$

When $s = 1$, IMisi then degenerates to PC (Fig. 1c), and we have $(b/c)^* = 1 - N < 0$, suggesting that cooperation is not favoured[17,21,29,30]. In fact, cooperation is not favoured in this case whenever $1 \leqslant s < (N-2)/(N-2d)$ since the critical benefit-to-cost ratio $(b/c)^*$ is negative. When $(N-2)/(N-2d) < s \leqslant d$, the critical ratio becomes positive and decreases as $s$ grows, suggesting more social information favours the evolution of cooperation. When $s = d$, meaning individuals have information about all neighbours, we recover the traditional IM rule[14], and $\lim_{N \to \infty}(b/c)^* = d + 2$, as reported previously[17]. In addition, heterogeneous networks also support these qualitative findings (see Methods).

When an individual's own information dominates ($\theta \to 1$), we obtain $\lim_{\theta \to 1}(b/c)^* = 1 - N$, which is independent of $s$. Interestingly, for well-mixed populations ($d = N - 1$), we get the same critical benefit-to-cost ratio $(b/c)^* = 1 - N$ from Eq. (4). This implies that when individuals depend almost exclusively on their own information to update strategies ($\theta \to 1$), the evolution of cooperation on graphs under IMisi resembles that of a well-mixed population. This finding echoes those obtained from aspiration-based update rules where individuals rely on only their own information for strategy updating[21,37–39].

To systematically explore the effect of personal information on the evolution of cooperation, we plot $(b/c)^*$ as a function of $\theta$ in Fig. 2c. As a general trend, we find that increasing the relative importance of an individual's own information ($\theta$) leads to an increase in $(b/c)^*$, suggesting that personal information is detrimental to the fixation of cooperation. Moreover, for fixed $\theta > 0$, we find that increasing $s$ can decrease $(b/c)^*$. Thus, when more social information is used during strategy updating, there is more room for cooperation to be favoured over defection. This is contrary to the intuition in previous studies[14], in which large

neighbourhoods impede cooperation. We find that $\theta = 0$ and $s > 1$ provide the best possible condition (lowest critical benefit-to-cost ratio) for the evolution of cooperation, suggesting that completely neglecting personal information best promotes the evolution of cooperation under pairwise interactions, and in this case, the amount of social information has no impact on the evolution of cooperation.

## Group social dilemmas

Even with complete social information, such as in standard DB, a remarkable property of pairwise interactions on regular networks is that the critical benefit-to-cost ratio depends on only the size, $N$, and the degree, $d$, of the network. This critical ratio was first derived for vertex-transitive graphs[24], which look the same from every vertex, and subsequently extended to regular graphs[36]. For IMisi, too, we find that pairwise interactions give a critical ratio that depends on just $N, d, \theta$, and $s$ (Eq. (4)). However, when considering group interactions, an individual's payoff can be affected by both one- and two-step (i.e., first- and second-order) neighbours, which suggests that clustering in the network plays a role in the evolution of cooperation. To simplify the expressions we report for group interactions, we now assume that the network is vertex-transitive of degree $d$, a slightly stronger notion of symmetry than regularity.

In the public goods game, we find that cooperators evolve whenever $r > r^*$, where

$$r^* = \frac{(d+1)^2\left(2\theta(N-1) + (1-\theta)\frac{d}{s}\frac{s-1}{d-1}(N-2)\right)}{2\theta(d+1)(N-d-1) + (1-\theta)\frac{d}{s}\frac{s-1}{d-1}\left(((d-1)\mathcal{C} + d + 3)N - 2(d+1)^2\right)}. \quad (6)$$

Here, $\mathcal{C}$ is the global clustering coefficient of the graph, which quantifies the overlap of first-order and second-order neighbours (for an explicit expression, see Methods). As shown in Fig. 3i, the critical multiplication factor, $r^*$, decreases as the clustering coefficient $\mathcal{C}$ increases, which means that highly clustered network structures generally promote the evolution of cooperation by reducing the barriers for selection to favour cooperators.

Although the critical ratio of Eq. (6) looks quite different from that of Eq. (4), there are some notable qualitative similarities between the two kinds of interactions. For example, when the relative importance of personal information takes extreme values ($\theta \to 0$ or $\theta \to 1$), the

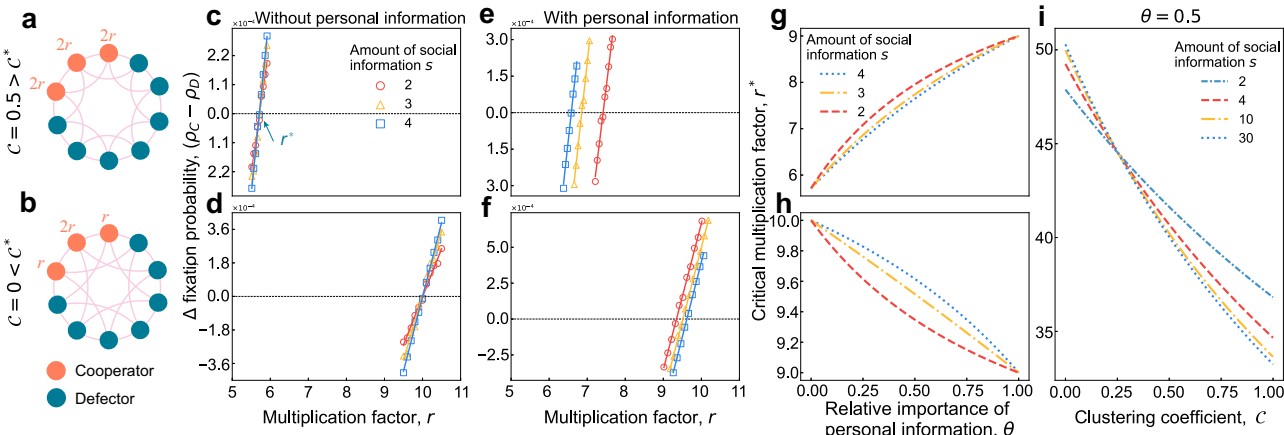

**Fig. 3 | Effects of incomplete information on the fixation of cooperation in group social dilemmas.** We consider two different regular networks with different clustering coefficients ($\mathcal{C}$). **a** On the graph with $\mathcal{C} = 0.5$, the payoffs of the public pools organised by cooperators are $2r$. **b** On the graph with $\mathcal{C} = 0$, the payoffs of the group organised by the cooperators on both sides decrease. We perform simulations on fixation probability difference $\rho_C - \rho_D$ as a function of multiplication factor $r$. Here markers are from numerical simulations and lines are from the corresponding linear curve fitting. When individuals ignore their own information ($\theta = 0$), $r^*$ is the same for different amounts of social information $s$ (**c, d**). The arrow points to the value of $r^*$ derived theoretically under weak selection (Eq. (6)). When

individuals treat both kinds of information equally ($\theta = 1/(s+1)$), the small amount of social information $s$ makes $r^*$ larger for $\mathcal{C} = 0.5 > \mathcal{C}^*$ (**e**). The influence is totally reversed when $\mathcal{C} = 0 < \mathcal{C}^*$ (**f**). We draw the critical $r^*$ as a function of the weight of personal information $\theta$ (Eq. (6)) for the networks in a and c. As $\theta$ goes up, $r^*$ increases when $\mathcal{C} = 0.5 > \mathcal{C}^*$ (**g**), and decreases when $\mathcal{C} = 0 < \mathcal{C}^*$ (**h**). The critical $r^*$ is a decreasing function of the clustering coefficient $\mathcal{C}$ for multi-player game when $\theta = 0.9$ (**i**). The curves converge when $\mathcal{C} = \mathcal{C}^*$, and then diverge, reversing the influence of the amount of social information $s$. Here, $c = 1$, and other parameters are the same as those in Fig. 2.

critical multiplication factor, $r^*$, is independent of $s$. Specifically, $\lim_{\theta \to 0} r^* = \frac{(d+1)^2(N-2)}{((d-1)\mathcal{C}+d+3)N-2(d+1)^2}$ and $\lim_{\theta \to 1} r^* = \frac{(d+1)(N-1)}{N-d-1}$. Despite these similarities, our findings for group interactions differ when $0 < \theta < 1$. Indeed, we find that there exists a critical threshold for the clustering coefficient,

$$\mathcal{C}^* = \frac{d^2 + d + 2 - 2N}{(N-1)(d-1)}, \quad (7)$$

which satisfies both $\partial r^*/\partial s < 0$ if and only if $\mathcal{C} > \mathcal{C}^*$ and $\partial r^*/\partial \theta > 0$ if and only if $\mathcal{C} > \mathcal{C}^*$. What this means is that, when $\mathcal{C} > \mathcal{C}^*$, the more social information that is used and the less that individuals weight their own information, the easier it is for cooperation to be favoured over defection (Fig. 3g). However, when $\mathcal{C} < \mathcal{C}^*$, the results are reversed, meaning that the more social information that is used and the less that individuals weight their own information, the harder it is for cooperation to be favoured over defection (Fig. 3h). Intuitively, if the network has a low level of clustering, then cooperative clusters are not robust and are easily exploited by defectors. In this case, it is better for cooperators to retain their strategy in order to increase the likelihood of survival during strategy competition to fill a vacancy.

To verify our theoretical results, we perform numerical simulations on two graphs with different clustering coefficients: $\mathcal{C} = 0.5 > \mathcal{C}^*$ (Fig. 3a) and $\mathcal{C} = 0 < \mathcal{C}^*$ (Fig. 3b). The effect of social information on cooperation is completely the opposite for large and small clustering coefficients (Fig. 3c, d, e, and f). When $\mathcal{C} = 0$, increasing social information and decreasing the weight of personal information increases $r^*$ and thus impedes the evolution of cooperation, but this effect is reversed when $\mathcal{C} = 0.5$. In addition to our explorations on regular graphs, we confirm that our findings are robust to heterogeneous network structures such as scale-free[40] and small-world networks[41] (see Methods and Supplementary Fig. S2). Moreover, we also numerically present the existence of the critical clustering coefficient determining the impact of social information in Supplementary Note 4 and Supplementary Fig. S8.

## The rate and range of competition induced by the IMisi update rule

Intuitively, the evolutionary dynamics generated by the IMisi rule can be understood as involving two competitive relationships. First, when an individual is selected to change its strategy, the focal individual competes with its neighbours to avoid imitation and retain its strategy. If it fails, the neighbours then compete to be the role model for imitation.

Regarding the evolutionary process, the spread of cooperation can be understood using a random walk on networks. Denote by $\bar{u}^{(n)}$ the expected payoff to an individual at the end of an $n$-step random walk from a cooperator (see Supplementary Note 2.3). Theoretically, we show that weak selection favours cooperators whenever

$$2\theta \frac{s}{d}\left(\bar{u}^{(0)} - \bar{u}^{(1)}\right) + (1-\theta)\frac{s-1}{d-1}\left(\bar{u}^{(0)} - \bar{u}^{(2)}\right) > 0. \quad (8)$$

The first term in this summation, weighted by $\theta$, is associated to competition between one-step neighbours. The weight $s/d$ is the probability of that a fixed neighbour is part of a focal individual's information set. The factor of 2 arises due to the two kinds of competition between one-step neighbours. The first occurs when a cooperator is chosen as the focal individual and competes to retain its strategy. The second occurs when a neighbour of the focal individual is a cooperator and is included in the focal individual's social information set. The remaining term in Eq. (8), weighted by $1 - \theta$, is associated to competition between two-step neighbours. Given a focal individual and a neighbour chosen for comparison, the probability that a fixed neighbour among the remaining nodes is part of the information set is $(s-1)/(d-1)$. Competition between these neighbours can be understood by placing a cooperator at one location and comparing the respective payoffs of the two players. Figure 4 illustrates the selection condition of Eq. (8).

It is difficult for cooperators to prevail in the competition with one-step neighbours. Specifically, the corresponding expected payoff of a focal cooperator is always less than the average of its random first-order neighbour, namely, $\bar{u}^{(0)} < \bar{u}^{(1)}$, because the first-order neighbours

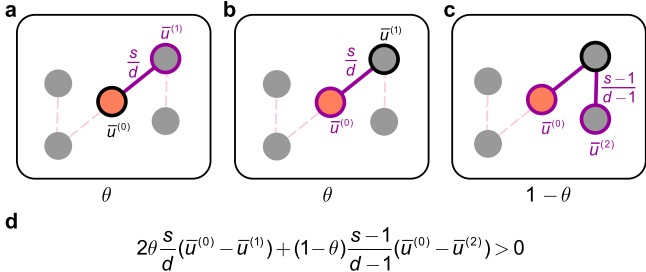

$$2\theta\frac{s}{d}(\overline{u}^{(0)} - \overline{u}^{(1)}) + (1-\theta)\frac{s-1}{d-1}(\overline{u}^{(0)} - \overline{u}^{(2)}) > 0$$

**Fig. 4 | Intuition about competition and the evolutionary success of cooperators.** The selection condition for cooperators to be favoured relative to defectors involves three kinds of competition, at two ranges. The individuals with black circles are the individuals who are changing strategy, and the individuals with purple circles linked by the purple lines are the individuals who are competing. **a** Conditioned on a cooperator (orange solid circle) being chosen as the focal individual (black circle) to evaluate its strategy, this cooperator competes with a first-order neighbour (purple circle) to retain its strategy. **b** Conditioned on a cooperator being a one-step neighbour of the focal individual (black circle), this cooperator competes to be a candidate (purple circle) for imitation. **c** Once the focal individual (black circle) decides to imitate some neighbour, a neighbouring cooperator competes with other neighbours (purple circle) to fill the vacancy. **d** Putting these three kinds of competition together, one obtains the selection condition reported in Eq. (8). Here, $\overline{u}^{(n)}$ is the expected payoff to an individual at the end of an $n$-step random walk from a cooperator.

of the focal cooperator always have a cooperative neighbour. However, competition with second-order neighbours is the key to the success of the evolution of cooperation. Success in such competition for a cooperator happens when its neighbour starts to cooperate. Thus, the payoff of the focal cooperator increases and an aggregation of cooperators forms. Hence, increasing the relative weight of competition with second-order neighbours can promote cooperation.

An individual's personal information and the number of social peers account for the range and rate of competition. Individuals may compete with first-order neighbours, second-order neighbours, or both (Fig. 4), depending on the information encoded in $\theta$ and $s$. As Eq. (8) shows, no competition occurs under neutral drift ($\theta=0, s=1$). If individuals neglect personal information and consider more than one piece of social information ($\theta=0$ and $s>1$), individuals compete only with their second-order neighbours for expansion, suggesting that the amount of social information has no impact on the critical value $(b/c)^*$ (Figs. 2a, 3c, d). Otherwise for $\theta>0$, if individuals only consult one piece of social information ($s=1$) or depend almost exclusively on their own information ($\theta \to 1$), individuals only compete with their first-order neighbours, which is harmful to the evolution of cooperation considering $\overline{u}^{(0)} < \overline{u}^{(1)}$. In other conditions with $\theta>0$ and $s>1$, individuals compete with both first-order and second-order neighbours. And increasing the amount of social information $s$ and decreasing the weight of personal information $\theta$ represent a larger relative weight of the competition with second-order neighbours, namely, $(1-\theta)(s-1)/(d-1)$ compared to that with first-order neighbours ($2\theta s/d$). This explains why using less personal information and more social information better facilitates cooperation.

Our findings shed light on the famous perplexing result[15,17] that regular networks promote the evolution of cooperation under DB but not under PC. Only competing with first-order neighbours leads to the finding that PC fails to promote cooperation. In contrast, it is possible for the expected payoff of a focal cooperator to exceed that of a random second-order neighbour under DB as long as $b/c$ is above a threshold. The nature of this threshold, as it depends on the amount of social information, the relative weightings, and the network structure, is captured by Eq. (4) in donation games and by Eq. (6) in public goods games.

## Heterogeneity in external information

So far, we have explored the scenario in which different individuals use the same amount of external social information (the same value of $s$). Considering that different individuals may have different abilities for collecting and processing social information, we next consider the scenario of heterogeneous social information. Let $s_i$ denote the number of neighbours that individual $i$ selects at random for comparison. We compare three distributions for $s_i$ (homogeneous, uniform, and Gaussian) in a population of size $N=100$ and degree $d=5$. Let $n(s)$ be the number of individuals having $s_i=s$. The homogeneous distribution fixes $s$ at 3 for all individuals, i.e. $n(3)=100$. For the uniform distribution, we use $n(1)=n(2)=n(3)=n(4)=n(5)=20$. For the Gaussian distribution, we use $n(3)=48, n(2)=n(4)=20$, and $n(1)=n(5)=6$. In each case, the mean of $s_i$ is 3.

When individuals do not take their own information into account during strategy updating ($\theta=0$), we find that the critical benefit-to-cost ratio holds the same for different distributions of $s_i$ (Fig. 5a, b). This means that, when $\theta=0$, heterogeneity in social information over different individuals does not qualitatively alter our results obtained under the homogeneous distribution. However, if individuals instead consider their own information for strategy updating, heterogeneity of social information usage generally hinders the fixation of cooperation (Fig. 5c, d).

These results again highlight the role that an individual's own information plays in the evolution of cooperation: it acts as a switch. When personal information is neglected during strategy updating, heterogeneity in the usage of social information has no impact on the evolution of cooperation; whenever personal information is considered, such heterogeneity generally inhibits the evolution of cooperation. This switching effect can be intuitively and approximately explained by our previous theoretical analysis. When $\theta=0$, we have shown that the amount of social information used does not affect $(b/c)^*$. When $\theta=1/(s+1)$, we see that

$$(b/c)^* > 0, \partial(b/c)^*/\partial s < 0, \text{ and } \partial^2(b/c)^*/\partial s^2 > 0, \quad (9)$$

whenever $1 \leqslant (N-2)/(N-2d) < s \leqslant d$ (see Eq. (5)). As a result, the rate at which $(b/c)^*$ decreases will slow down as $s$ increases, which explains the inhibitory effect of heterogeneous usage of social information: when the number of neighbours $s_i$ that an individual consults deviates from the average value $\overline{s}$, individuals with $s_i < \overline{s}$ will induce an inhibitory effect on cooperators, and it cannot be counterbalanced by the positive effects led by those individuals with $s_i > \overline{s}$. This also explains why we observe that the homogeneous distribution is superior to uniform and Gaussian distributions: a smaller standard error (Fig. 5) indicates there are fewer individuals using $s_i \neq \overline{s}$.

## Discussion

Many classical evolutionary processes in networked systems can be interpreted as being intelligent, cultural and arising from imitation dynamics. Although these processes are abstractions of reality and cannot capture every aspect of the intricacies of intelligent animal and human behaviour, they are often amenable to mathematical analysis, which yields important insights into how traits spread over systems. In an overwhelming majority of these models, the imitation mechanism lies on an extreme end of the spectrum, involving either complete or very limited external information. Furthermore, they frequently assume that individuals interact with only first-order neighbours. In this study, we have considered a natural family of parametrised update rules, which includes classical imitation processes as special cases. We have analysed this model in terms of general payoff relationships, which allows for the study of traditional social dilemmas with first-order neighbours, like the donation game, as well as group interactions with individuals farther afield, including public goods games. Our framework can be easily extended to investigate imitation dynamics

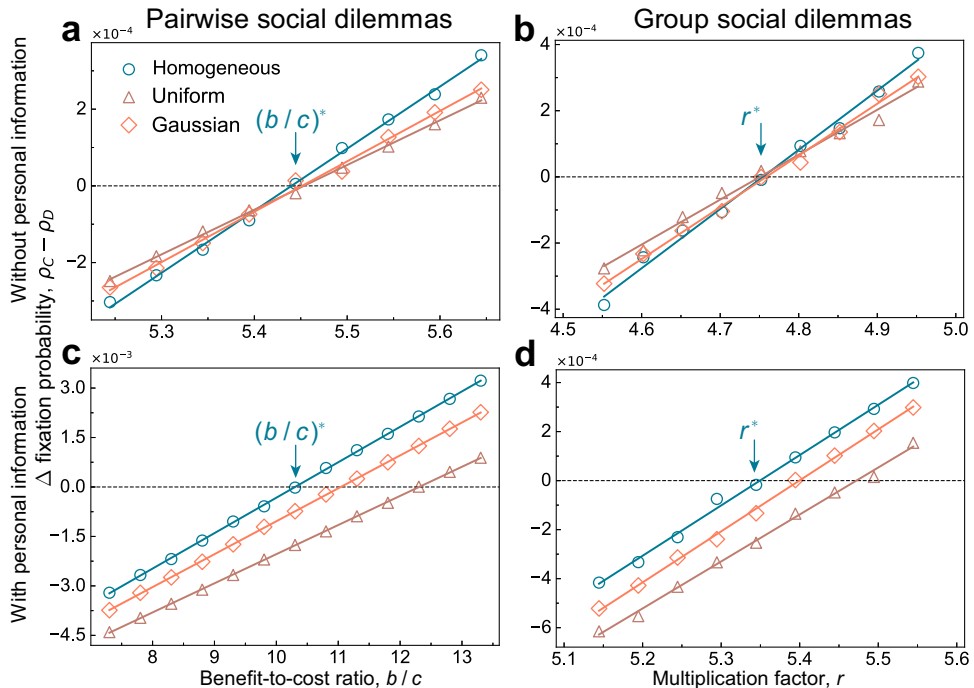

**Fig. 5 | Effects of heterogeneous social information on the fixation of cooperation.** For three different distributions of external social information (homogeneous, uniform, and Gaussian), we present the fixation probability difference $\rho_C - \rho_D$ for pairwise and group social dilemmas on regular graphs. Here, markers are from numerical simulations and lines are from the corresponding linear curve fitting. The vertical arrows point to the values of $(b/c)^*$ and $r^*$ derived theoretically. When personal information is not considered ($\theta = 0$), we find that information

heterogeneity does not change the critical values (i.e., $(b/c)^*$ and $r^*$ as shown in Figs. 2 and 3) over different distributions for pairwise (**a**) and group (**b**) interactions. When an individual's personal information is taken into account, the results change (**c**, **d**), showing that the homogeneous distribution generates the smallest value of $(b/c)^*$ and $r^*$. The mean for the homogeneous, uniform, and Gaussian distributions of social information is 3, and the standard error is 0, 1.41, and 0.90, respectively.

based on nonlinear multi-player games[42,43] and general group interactions[44]. In addition, how different types of personal and external information affect the evolution of cooperation in heterogeneous networks, from a theoretical viewpoint, deserves further investigations. Exploring the independent impact of personal information and social information is also worthwhile[45].

For the prosperity of altruistic behaviour in donation games, individuals should ignore personal information and rely more on social peers for comparison. The situation is more nuanced in public goods games, as clustering in the network plays a greater role. There, another critical threshold appears for the clustering coefficient. Above this threshold, cooperation is more easily favoured by weighting one's own success less and using more neighbours for comparison. Below this threshold, these findings are flipped. In fact, the appearance of clustering coefficients is interesting in and of itself, even when restricted to a classical mechanism like DB. Clustering is absent from the analysis of donation games altogether, and our results show that the critical multiplication factor in public goods games is a monotonically decreasing function of the clustering coefficient, reflecting the fact that cooperation in these games is favoured most when there is significant overlap between first- and second-order neighbours. The differing results between pairwise and group interactions are mainly due to the sparsity of connections: with sparse connections, defectors easily exploit cooperators through group interactions even when they are inside cooperative clusters. In such settings, it is better for cooperators to weight their own success more when deciding whether to imitate a neighbour.

The first-order competition for resisting strategy change is an instance of the so-called "status quo bias" in economics and psychology[46,47]. People tend to be inclined to keep their present behaviour, especially when they are successful. Our results show that,

for the emergence of cooperation, such behaviour is not necessarily conducive to the well-being of the community. Learning from better-performing individuals, represented by second-order competition, is often a more efficient way to promote the spread of altruism. And it has been shown that such a process plays an important role in human decision-making[48–51]. Our study provides possible intuition for how coupling this inherent human psychological activity to incomplete social information influences the emergence of collective cooperation.

The IMisi rule and its selection condition, Eq. (8), raise the question of relationships to classical imitation rules on weighted graphs. For example, under IM dynamics, when an individual itself is weighted by $\theta_s$ and neighbours are weighted by $\theta_n$, the probability that $j$ imitates $i \neq j$ is $\theta_n F_j h_{ji} / (\theta_n \sum_{k=1}^{N} F_k h_{ki} + \theta_s F_i)$, where $h_{ij}$ stands for the weight of the edge between $i$ and $j$. Although such an update rule is evidently distinct from that of Eq. (3), it is not immediately obvious that this is so under the assumption of weak selection. By way of analogy, stochastic payoff schemes can be reduced to deterministic models (i.e. in expectation) under weak selection[30]. In the present model, intriguingly, one cannot generally find weights $\theta_s$ and $\theta_n$ such that the weak-selection dynamics generated match those of Eq. (3) (see Methods). Therefore, generically, IMisi constitutes a new class of imitation mechanisms.

From a modelling perspective, our approach departs from the standard paradigm of fixing the update rule and varying the system structure. Instead, we fix a class of (regular) networks and study the effects of changing the parameters of the update rule on the evolution of cooperation. This approach is similar in spirit to that of Grafen and Archetti[52], who studied the effects of the range of density dependence on the evolution of altruism, which in turn illuminated why update rules with global competition for

reproduction do not favour cooperation while others, with involving more localised competition, can. Recently, there has also been a focus on classifying the pertinent update rules for (meta-) populations of fixed structure[53]. Such update rules can involve several steps (birth, death, and migration at various levels), and it is an open problem how each of these steps affects the evolution of density-dependent behaviours. Although the specific motivation for our study is quite different from these earlier works, it fits into the theme of understanding how the microscopic details of reproduction and survival affect the evolutionary dynamics of a population, which will continue to be an important task going forward.

## Methods

### Notation and payoff calculation

The population consists of $N$ individuals, and its structure is represented by a $d$-regular graph, $G$. The state of the population can be represented by a binary vector, $\mathbf{x} \in \{0,1\}^N$, where $x_i = 1$ indicates that individual $i$ is a cooperator and $x_i = 0$ means a defector. Let us consider random walks on $G$ in discrete time. For a random walk on the regular graph $G$, the probability of a one-step walk from node $i$ to node $j$ is $p_{ij} = 1/d$ if they are connected; otherwise, $p_{ij} = 0$. We denote by $p_{ij}^{(m)}$ the probability of going from $i$ to $j$ in an $m$-step random walk. Since the graph is regular, the unique stationary distribution places weight $\lim_{m \to \infty} p_{ij}^{(m)} = 1/N$ on node $j$. Let $\bar{u}_i^{(m)}$ be the expected average payoff of an individual at the end of an $m$-step random walk from individual $i$. Under pairwise interactions in the donation game, a cooperator pays a cost $c$ to offer its opponent a benefit $b$ and a defector pays nothing and provides no benefit. Thus, the average payoff is

$$\bar{u}_i^{(m)} = -cx_i^{(m)} + bx_i^{(m+1)}, \tag{10}$$

where $x_i^{(m)} = \sum_{j \in G} p_{ij}^{(m)} x_j$ represents the probability that an individual at the end of an $m$-step random walk from individual $i$ is a cooperator. Intuitively, the first term in Eq. (10) represents the expected cost incurred for the individuals $m$-step away if they cooperate. The second term represents the benefits that these individuals receive when their neighbours cooperate. For group interactions, a cooperator pays a cost $c$ in each game, and the total cost from cooperators is then enhanced by a multiplication factor $r$ and divided among all members of the group (i.e. the focal individual who organises the game and its $d$ neighbours). Without loss of generality, here we set $c = 1$. According to the definition and the detailed derivation presented in Supplementary Note 2.3, we have

$$\bar{u}_i^{(m)} = r\frac{d^2}{d+1}x_i^{(m+2)} + r\frac{2d}{d+1}x_i^{(m+1)} + \left[r\frac{1}{d+1} - (d+1)\right]x_i^{(m)}. \tag{11}$$

For a detailed derivation of Eq. (11), please refer to Supplementary Note 2.3.

### General condition for the success of cooperators

Let $D(\mathbf{x})$ be the expected instantaneous rate of change in the frequency of strategy C. We have $D(\mathbf{x}) = \sum_{i \in G} x_i(b_i(\mathbf{x}) - d_i(\mathbf{x}))$, where $b_i(\mathbf{x})$ is the probability that $i$ replaces one of its neighbours and $d_i(\mathbf{x})$ is the probability that it is replaced by its neighbours[18]. Intuitively, $D(\mathbf{x}) > 0$ represents a net increase of cooperators. Under neutral drift ($\delta = 0$), $D(\mathbf{x}) = 0$. Thus, we have $D(\mathbf{x}) = \delta \partial D(\mathbf{x})/\partial \delta + \mathcal{O}(\delta^2)$. Consequently, under weak selection ($0 < \delta \ll 1$), the condition for cooperation to be favoured over defection is

$$\left\langle \frac{\partial}{\partial \delta} D(\mathbf{x}) \right\rangle^\circ = \left\langle \frac{\partial}{\partial \delta} \sum_{i \in G} x_i(b_i(\mathbf{x}) - d_i(\mathbf{x})) \right\rangle^\circ > 0, \tag{12}$$

where $\langle \cdot \rangle^\circ$ represents the expectation over states arising under neutral drift. Intuitively, Eq. (12) guarantees that the average difference between the birth and death rates for cooperators ($x_i = 1$) is larger than zero, indicating a net increase in the population of cooperators.

Based on Eq. (12), we derive a general condition for cooperation to be favoured over defection, which reads

$$\left\langle \sum_{i \in G} \frac{x_i}{N}\left[2\theta\frac{s}{d}(\bar{u}_i^{(0)} - \bar{u}_i^{(1)}) + (1-\theta)\frac{s-1}{d-1}(\bar{u}_i^{(0)} - \bar{u}_i^{(2)})\right]\right\rangle^\circ > 0. \tag{13}$$

The equation makes sense when $x_i = 1$, which means that the individual $i$ is a cooperator. $\bar{u}_i^{(0)}$ is the payoff of the cooperator, and $\bar{u}_i^{(m)}$ ($m \geq 1$) is the payoff of a random $m$th-order neighbour of the cooperator. As demonstrated in Eq. (12), to ensure that the expected instantaneous rate of change of cooperators remains greater than zero, it is imperative for the average payoffs of cooperators to exceed those of their surrounding competitors. Eq. (13) states that for cooperators to be favoured, the net result of the combination of three types of competition that a cooperator engages in should be positive: (i) competition with a random first-order neighbour for not being replaced, $\bar{u}_i^{(0)} - \bar{u}_i^{(1)}$, occurring with weight $\theta$; (ii) competition with a random first-order neighbour to replace it, $\bar{u}_i^{(0)} - \bar{u}_i^{(1)}$, occurring with weight $\theta$; and (iii) competition with one of the second-order neighbours for finally replacing its first-order neighbours, $\bar{u}_i^{(0)} - \bar{u}_i^{(2)}$ with weight $(1 - \theta)$. Here, $(s-1)/(d-1)$ is the probability for a second-order neighbour to be randomly selected to participate in the competition, given that the cooperator has already been selected.

### Condition for success under pairwise and group interactions

To calculate condition (13), we introduce the coalescing random walk, which is a collection of random walks that step independently until two walks meet[18]. Let $\tau_{ij}$ denote the expected coalescence time between $i$ and $j$ under the discrete-time coalescing random walk. Suppose $i$ and $j$ are the two ends of a random walk of length $m$. Analogous to other imitation-based update rules, we have $\tau_{ii} = 0$ and

$$\tau_{ij} = \frac{1}{1-\theta} + \frac{1}{2}\sum_{k=1}^{N} p_{ik}^{(1)}\tau_{kj} + \frac{1}{2}\sum_{k=1}^{N} p_{jk}^{(1)}\tau_{ik}, \tag{14}$$

for $i \neq j$. We let $\tau^{(m)} = \sum_{i,j \in G} p_{ij}^{(m)} \tau_{ij}/N$, which represents the expectation of $\tau_{ij}$ over all possible choices of $i$ and $j$ in the stationary distribution of the random walk. According to a previous study[18], for $m_1, m_2 \geq 0$, we have

$$\left\langle \sum_{i \in G} \frac{1}{N} x_i \cdot (x_i^{(m_1)} - x_i^{(m_2)}) \right\rangle^\circ = \frac{\tau^{(m_2)} - \tau^{(m_1)}}{2N}. \tag{15}$$

Letting $\tau_{ii}^+ = 1/(1-\theta) + \sum_{j \in G} p_{ij}\tau_{ij}$ be the expected remeeting time in the discrete-time random walk, we have

$$\tau^{(m+1)} - \tau^{(m)} = \sum_{i \in G} \frac{1}{N} p_{ii}^{(m)}\tau_{ii}^+ - \frac{1}{1-\theta}, \tag{16}$$

where $p_{ii}^{(m)}$ denotes the probability that an $m$-step random walk terminates at its starting position $i$. In particular, for regular graphs, we have $\tau_{ii}^+ = N/(1-\theta)$ and $p_{ii}^{(m)} = p^{(m)}$ for all $i \in G$[18]. Now, we have that for regular graphs with degree $d$,

$$p^{(1)} = 0, \quad p^{(2)} = \frac{1}{d}, \quad p^{(3)} = \frac{d-1}{d^2}\mathcal{C}. \tag{17}$$

Substituting Eqs. (15), (16), and (17) to Eq. (12), we have that for pairwise interactions

$$\left\langle \frac{\partial}{\partial \delta} D \right\rangle^\circ = \frac{1}{2N} \left\{ \left( 2\theta + (1-\theta)\frac{(s-1)d}{s(d-1)} \right)[b(Np^{(1)}-1) - c(Np^{(0)}-1)] \right.$$
$$+ (1-\theta)\frac{(s-1)d}{s(d-1)}[b(Np^{(2)}-1) - c(Np^{(1)}-1)] \Big\}$$
$$= \frac{1}{2N} \left\{ b\left[ N\left( (1-\theta)\frac{(s-1)}{s(d-1)} \right) - 2(1-\theta)\frac{(s-1)d}{s(d-1)} - 2\theta \right] \right.$$
$$\left. -c\left[ N\left( (1-\theta)\frac{(s-1)d}{s(d-1)} + 2\theta \right) - 2(1-\theta)\frac{(s-1)d}{s(d-1)} - 2\theta \right] \right\}.$$
(18)

Similarly, for group interactions, we have

$$\left\langle \frac{\partial}{\partial \delta} D \right\rangle^\circ = \frac{(d+1)}{2N} \left\{ r\left[ N\left( (\mathcal{C}(d-1)+2)(1-\theta)\frac{(s-1)d}{s(d-1)} \right) \right. \right.$$
$$+ (d+1)\left( (1-\theta)\frac{(s-1)d}{s(d-1)} + 2\theta \right) \Big)/(d+1)^2$$
$$-2(1-\theta)\frac{(s-1)d}{s(d-1)} - 2\theta \Big]$$
$$\left. -N\left( (1-\theta)\frac{(s-1)d}{s(d-1)} + 2\theta \right) + 2(1-\theta)\frac{(s-1)d}{s(d-1)} + 2\theta \right\}.$$
(19)

Solving $\left\langle \frac{\partial}{\partial \delta} D \right\rangle^\circ > 0$, we recover conditions (4) and (6) for the success of cooperators.

### Simulations on heterogeneous networks

We performed simulations under different amounts of social information on two well-known classes of heterogeneous networks: small-world networks[41] and Barabási-Albert networks[40]. For both networks, the average degrees are set to $\bar{d} = 6$, and we take the minimum degree of each network to be 3. Due to degree heterogeneity, the number of neighbours may vary for different individuals. We perform the simulations for $1 \le s \le 3$ and for the cases where individuals know all the social information. Two relative weights of personal information are considered, namely $\theta = 0$ and $\theta = 1/(s+1)$.

When $s > 1$ and $\theta = 0$, the critical benefit-to-cost ratio $(b/c)^*$ is the same for various amounts of social information (Supplementary Figs. S1a, c, and S2a, c). This indicates that if an individual's personal information is neglected, the amount of social information has no impact on the evolution of cooperation. When $\theta = 1/(s+1)$, the critical ratio $(b/c)^*$ decreases as $s$ increases, meaning cooperation is promoted (Supplementary Figs. S1b, d and S2b, d). These results are consistent with our findings on regular graphs.

Compared with our results on regular graphs, heterogeneity does affect the evolution of cooperation. Under donation game, for regular graphs, $(b/c)^* = 6.68$ when $\theta = 0$, $N = 100$, and $d = 6$. Barabási-Albert networks have inhibitory effects on the evolution of cooperation ($(b/c)^* \approx 7.2$). But small-world networks can slightly promote cooperation ($(b/c)^* \approx 6.4$). However, this effect of small-world networks is not strong at $\theta = 1/(s+1)$. In public goods game, small-world networks have larger clustering coefficients[41], which may be the reason why these kinds of networks better facilitate the evolution of cooperation. Nevertheless, when $\theta = 1/(s+1)$, the inhibitory effect with small amounts of social information on small-world networks is more profound. When all social information is known, small-world networks are better for the evolution of cooperation. When $s = 1$, the critical value $(b/c)^*$ of the small-world networks grows rapidly, surpassing the corresponding value of Barabási-Albert networks.

We also investigate the impact of heterogeneous levels of social information on cooperation in Barabási-Albert networks[40]. The distribution of social information $s_i$ is the same as those in Fig. 5. Our findings reveal that the conclusions remain unchanged on heterogeneous networks (Supplementary Fig. S3). Specifically, when $\theta = 0$,

the critical benefit-to-cost ratio stays the same for various distributions of $s_i$. When $\theta > 0$, homogeneous distributions of social information predominantly foster cooperation, while the heterogeneity of social information tends to impede the fixation of cooperation.

### Relationship to classical imitation dynamics

Since the imitation rule we consider involves choosing $s$ model individuals uniformly from all $d$ neighbours, a natural question to ask is whether the dynamics are equivalent to those of classical imitation dynamics ("IM")[14] on weighted graphs, with one weight $\theta_s$ corresponding to the individual itself and another $\theta_n$ corresponding to neighbours. That all neighbours correspond to the same weight, $\theta_n$, arises from the assumption that social information sets are sampled uniformly under IMisi. The weight of the edge between $i$ and $j$ is $h_{ij}$. For such a process, the probability that $i$ imitates $j$'s strategy in state $\mathbf{x} \in \{0,1\}^N$ is

$$\frac{1}{N} \frac{\theta_n F_j(\mathbf{x}) h_{ji}}{\theta_n \sum_{k=1}^N F_k(\mathbf{x}) h_{ki} + \theta_s F_i(\mathbf{x})},$$
(20)

and the probability that $i$ keeps its own strategy is

$$\frac{1}{N} \frac{\theta_s F_i(\mathbf{x})}{\theta_n \sum_{k=1}^N F_k(\mathbf{x}) h_{ki} + \theta_s F_i(\mathbf{x})}.$$
(21)

For general selection intensity, $\delta$, this update rule is clearly different from the one defined by Eq. (3), but we can still ask about weak-selection dynamics. By scaling $\theta_s$ and $\theta_n$, we may assume that $\theta_n d + \theta_s = 1$. Differentiating both transmission probabilities with respect to $\delta$ at $\delta = 0$ and searching for appropriate weights $\theta_s$ and $\theta_n$ ($= (1-\theta_s)/d$) such that the two derivatives are equal, we find that $\theta(1-\theta) = \theta_s(1-\theta_s)$, $(1-\theta)^2 \frac{1}{s}\frac{s-1}{d-1} = (1-\theta_s)\theta_n$, and $(1-\theta)(1 - \frac{1}{s}(1-\theta)) = (1-\theta_s)(1-\theta_n)$. The first equation implies that $\theta_s = \theta$ or $\theta_s = 1-\theta$, and in either of these cases the second and third equations are equivalent. If $\theta_s = \theta$, then the second equation requires $s = d$. If $\theta_s = 1-\theta$, then the second equation requires $\theta = \frac{\sqrt{\frac{s-1}{d-1}}}{\sqrt{\frac{s-1}{d-1}} + \sqrt{\frac{s}{d}}}$. Thus, generically, IMisi dynamics are not equivalent to IM dynamics on a weighted graph, even under weak selection.

### Reporting summary

Further information on research design is available in the Nature Portfolio Reporting Summary linked to this article.

## Data availability

All data generated or analysed during this study are included within the paper and its supplementary information files.

## Code availability

All numerical calculations and computational simulations were performed in Julia 1.4.1. All data analyses were performed in Python 3.9.12. All codes have been deposited into the publicly available repository at https://zenodo.org/record/8430355.

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

## Acknowledgements

We thank Yao Meng, Zhenglong Tian and other members of the Li Lab for helpful comments and discussions. X.W. and A.L. gratefully acknowledge the support from the National Key Research and Development Program of China under Grant No. 2022YFA1008400, the National Natural Science Foundation of China (NSFC) under Grant No. 62173004,

the Beijing Nova Program, China, under Grant No. Z211100002121105. L.Z. is supported by the National Key Research and Development Program of China under Grant No. 2022YFA1004702, the National Natural Science Foundation of China under Grant No. 62303049, and the Beijing Institute of Technology Research Fund Program for Young Scholars.

## Author contributions

X.W., L.Z. and A.L. conceived and designed the research; All authors performed the research and analysed the results; X.W. and L.Z wrote the computer codes for numerical simulations; X.W. performed theoretical analyses under the help of L.Z., A.M. and A.L.; All authors wrote the paper and approved the submission.

## Competing interests

The authors declare no competing interests.
