## [Peer Review File · Nature Communications]

Reviewers' Comments:

Reviewer #1:

Remarks to the Author:

This study offers a novel and thought-provoking perspective on the role of imitation dynamics within structured populations, a topic that I found deeply engaging. Among the most striking features of this rigorously conducted research is the introduction of a new general model of imitation dynamics, accounting for incomplete social information; a feature that is often neglected in previous models. The comprehensive analytical and simulation results presented in the study are a testament to the potential of this approach in bridging knowledge gaps and unifying classical rules. Also, to my knowledge, the literature review offers a comprehensive overview of previous significant studies published. The topic is highly relevant for the design of novel online experiments involving human subjects.

The application of this model to the realm of pairwise social dilemmas has yielded significant and insightful results, including the finding that cooperation is most promoted when personal information is neglected. Moreover, the manuscript investigates various critical factors, such as the rate and range of competition induced by imitation dynamics, heterogeneity in social information levels, and conditions for the success of cooperators under different interactions. This depth of exploration, along with the use of simulations on heterogeneous graphs and the connections drawn to classical imitation dynamics, solidifies the paper's comprehensive approach.

Although the study does leave room for deeper clarification on how different types of personal and social information affect the evolution of cooperation in heterogeneous agent populations, it still offers valuable findings. Particularly, the idea that cooperation in low-clustering networks is better facilitated with more personal and less social information presents an intriguing contribution to the existing body of knowledge.

Overall, I found the manuscript an enjoyable read. I believe it stands strong as it is, with no need for further structural or results enhancements to warrant consideration for publication. Its groundbreaking approach, original model, and considerable findings are poised to make significant impacts on the field. I strongly recommend the publication of this study.

Reviewer #2:

Remarks to the Author:

The evolution of cooperation in networked populations has deserved the attention of a wide multidisciplinary community and several previous key results have been published, including in Nature Communications. In this paper the authors continue this tradition by elaborating on the information sources — social or personal — required for cooperation to prevail on networks. The authors present a general model where two key parameters (θ and s) control the mode of information to update strategies on networks, interpolating between death-birth, imitation, and pairwise-comparison modes. The paper then proceeds to describe how the two key parameters impact the critical benefit to cost ratio (in a donation game) and multiplication factor (in a public goods game) that leads the fixation probability of cooperators to be higher than the fixation probability of defectors. The theoretical analysis conducted is validated with numerical simulations on heterogeneous networks.

This paper shows that cooperation increasingly stable if individuals ignore personal information and rely more on social information. This relation is not linear, nonetheless: in group social dilemmas on networks with high clustering, relying on personal information leads to higher levels of cooperation. I found the conclusions in this paper quite thought-provoking and of high relevance to academics interested in the evolution of cooperation. The model presented is in my opinion well formulated and elegant. The analysis appears sound to me, although the intuition for the methods used and the

motivation for some simulations should be clarified (see below).

*Some optional recommendations to revise:

- The presentation of the methods in the main text can be improved to be accessible for a wide audience: 1) the connection between long random walks (large m) and the calculation of local payoffs on networks should be clarified; 2) Extra intuition should be given for the conditions in equations 12, 13 and the average that is computed should be clarified; is that the average over network realisations? Over time? Over different initial conditions? Overall, although the method used to compute the relative advantage of cooperators on networks is presented in prior papers, this manuscript should be self-contained, and more intuition should be provided for this approach that involves random walks and comparison with dynamics under weak selection.

- The core results of this paper are analytical and numerical simulations are used to mainly validate heterogeneity. The authors could justify why Figure 2 (A and B) presents simulation results and not the b/c ratio computed analytically. What is really being relaxed here, compared with the analytic model, that justifies presenting simulations? The authors should also clarify in the caption the meaning of the vertical arrows plotted in panel A and B.

- In Figure 3 only two results of global clustering are tested. As a result, in my opinion Figure 3 does not fully stress that there is a critical clustering coefficient to determine the impact of social information and that this critical value is the one derived analytically. One alternative to extend the analysis is to test the obtained critical b/c ratio for several realizations of the Watts-Strogatz small-world networks with an increasing p (probability of rewiring). This allows to interpolate between high clustering (0.5) and very low clustering coefficient (for $p=1$).

*Minor points:

-line 47: nature section: natural selection

-line 95: the authors can present here the range of r that guarantees that the game played is a social dilemma

-line 133: derive intuition: to derive an intuition for?

-line 170: depends on only the size N , and the degree, d of the network.

-line 296: minimize: disregard? ignore?

-line 312: I wonder whether this can also be an example of "status quo bias"

-methods: In the beginning of the methods section the authors should recall the meaning of b and c , as well as the relevance of long random walks for the computational of an individual's payoff.

-code availability: Regarding code availability the authors report the IDE that was used to code the analysis performed and do not detail the custom data that was created. It would be more relevant, to replicate the results, to report on the version of Julia that was used and, eventually, the computational resources required (such as time and memory) to replicate the findings.

-Figure 3: some labels are impossible to read given the small font.

Response to Reviewer #1

This study offers a novel and thought-provoking perspective on the role of imitation dynamics within structured populations, a topic that I found deeply engaging. Among the most striking features of this rigorously conducted research is the introduction of a new general model of imitation dynamics, accounting for incomplete social information; a feature that is often neglected in previous models. The comprehensive analytical and simulation results presented in the study are a testament to the potential of this approach in bridging knowledge gaps and unifying classical rules. Also, to my knowledge, the literature review offers a comprehensive overview of previous significant studies published. The topic is highly relevant for the design of novel online experiments involving human subjects.

The application of this model to the realm of pairwise social dilemmas has yielded significant and insightful results, including the finding that cooperation is most promoted when personal information is neglected. Moreover, the manuscript investigates various critical factors, such as the rate and range of competition induced by imitation dynamics, heterogeneity in social information levels, and conditions for the success of cooperators under different interactions. This depth of exploration, along with the use of simulations on heterogeneous graphs and the connections drawn to classical imitation dynamics, solidifies the paper's comprehensive approach.

We thank Reviewer #1 for his/her positive summary of our work, and for pointing out that we introduced a new general model of imitation dynamics accounting for incomplete social information that is often neglected in previous models.

Although the study does leave room for deeper clarification on how different types of personal and social information affect the evolution of cooperation in heterogenous agent populations, it still offers valuable findings.

We thank Reviewer #1 for pointing this out. We agree that this is indeed a topic worthy of investigation. To delve into this, in addition to different types of information, we further consider another dimension of heterogeneity in our model, namely, the heterogeneity in the number of social ties in the revised version. Our findings are consistent with the results presented in Figure 5 of the main text, which consider heterogeneous levels of social information but homogeneous social ties (Fig. R1, namely Supplementary Figure S3 in the revised Supplementary Information).

Indeed, we fully agree with Reviewer #1, and based on our findings and framework, we believe that how different types of personal and social information affect the evolution of cooperation in heterogeneous populations deserves further investigations, which is now pointed out in the revised Discussion section.

Figure R1: Effects of heterogeneous social information on the fixation of cooperation on Barabási-Albert networks. For three different distributions of social information (identical, uniform, and Gaussian), we present the fixation probability difference $\rho_C - \rho_D$ for pairwise and group social dilemmas on Barabási-Albert networks⁵³. The distributions are the same as Fig. 5 in the main text. Here, markers are from numerical simulations and lines are from the corresponding linear curve fitting. When self-information is not considered ($\theta = 0$), we find that information heterogeneity does not change the critical values (i.e., $(b/c)^*$ and r^*) over different distributions for pairwise (a) and group (b) interactions. When an individual's self-information is taken into account, the results change (c, d), showing that the homogeneous distribution generates the smallest value of $(b/c)^*$ and r^* . Note that these results are consistent with those in Fig. 5 which focuses on regular networks. Here, the parameters are $N = 100$, $\langle d \rangle = 10$, $\delta = 0.01$.

Particularly, the idea that cooperation in low-clustering networks is better facilitated with more personal and less social information presents an intriguing contribution to the existing body of knowledge.

We thank Reviewer #1 for pointing out the novel contribution of our work.

Overall, I found the manuscript an enjoyable read. I believe it stands strong as it is, with no need for further structural or results enhancements to warrant consideration for publication. Its groundbreaking approach, original model, and considerable findings are poised to make significant impacts on the field. I strongly recommend the publication of this study.

We thank Reviewer #1 again for the positive and careful review.

Response to Reviewer #2

The evolution of cooperation in networked populations has deserved the attention of a wide multidisciplinary community and several previous key results have been published, including in Nature Communications. In this paper the authors continue this tradition by elaborating on the information sources — social or personal — required for cooperation to prevail on networks. The authors present a general model where two key parameters (θ and s) control the mode of information to update strategies on networks, interpolating between death-birth, imitation, and pairwise-comparison modes. The paper then proceeds to describe how the two key parameters impact the critical benefit to cost ratio (in a donation game) and multiplication factor (in a public goods game) that leads the fixation probability of cooperators to be higher than the fixation probability of defectors. The theoretical analysis conducted is validated with numerical simulations on heterogeneous networks.

This paper shows that cooperation increasingly stable if individuals ignore personal information and rely more on social information. This relation is not linear, nonetheless: in group social dilemmas on networks with high clustering, relying on personal information leads to higher levels of cooperation. I found the conclusions in this paper quite thought-provoking and of high relevance to academics interested in the evolution of cooperation. The model presented is in my opinion well formulated and elegant. The analysis appears sound to me, although the intuition for the methods used and the motivation for some simulations should be clarified (see below).

We thank Reviewer #2 for reviewing our manuscript and pointing out that our conclusions are quite thought-provoking and the model is well formulated and elegant. Below we provide point-by-point responses to each of the reviewer's comments and suggestions.

*Some optional recommendations to revise:

- The presentation of the methods in the main text can be improved to be accessible for a wide audience: 1) the connection between long random walks (large m) and the calculation of local payoffs on networks should be clarified; 2) Extra intuition should be given for the conditions in equations 12, 13 and the average that is computed should be clarified; is that the average over network realisations? Over time? Over different initial conditions? Overall, although the method used to compute the relative advantage of cooperators on networks is presented in prior papers, this manuscript should be self-contained, and more intuition should be provided for this approach that involves random walks and comparison with dynamics under weak selection.

Very good point. We appreciate the suggestions provided by Reviewer #2. These recommendations enhance the comprehensiveness and readability of our manuscript.

1) The local payoff for pairwise social dilemma is

$$\bar{u}_i^{(m)} = -c\bar{x}_i^{(m)} + b\bar{x}_i^{(m+1)}$$

Intuitively, the first term represents the expected cost incurred for the individuals m -step away if they cooperate. The second term represents the benefits that these individuals receive when their neighbours cooperate. Following your constructive comment, we have carefully clarified the connection in the revised version, and further explain the long derivation of the payoff for group social dilemma in Supplementary Information in detail.

2) The Eqs. (12) and (13) yield the condition for cooperation to be favoured over defection. Let $D(\mathbf{x}) = \sum_{i \in G} x_i (b_i(\mathbf{x}) - d_i(\mathbf{x}))$ be the expected instantaneous rate of change in the frequency of strategy C, where $b_i(\mathbf{x})$ is the probability that i replaces one of its neighbours and $d_i(\mathbf{x})$ is the probability that it is replaced by its neighbours. Generally, $D(\mathbf{x}) > 0$ represents a net increase of cooperators, and under neutral drift ($\delta = 0$), $D(\mathbf{x}) = 0$. Thus, we have $D(\mathbf{x}) = \delta \frac{\partial}{\partial \delta} D(\mathbf{x}) + O(\delta^2)$. The condition then becomes Eq. (12). Eq. (13) is the condition after substituting the payoff specifically. Intuitively, to ensure that the expected instantaneous rate of change of cooperators remains greater than zero, it is imperative for the average payoffs of cooperators to exceed those of their surrounding competitors. And the averaging process is conducted over the state that emerges under neutral drift ($\delta = 0$).

Motivated by the reviewer's comments, in the revised manuscript, we have given extra intuitions for the conditions in Eqs. (12) and (13), and clarified the process on how the average is computed over the state. Moreover, we also take this chance to present the intuitive meaning of each variable.

- The core results of this paper are analytical and numerical simulations are used to mainly validate heterogeneity. The authors could justify why Figure 2 (A and B) presents simulation results and not the b/c ratio computed analytically. What is really being relaxed here, compared with the analytic model, that justifies presenting simulations? The authors should also clarify in the caption the meaning of the vertical arrows plotted in panel A and B.

We thank Reviewer #2 for the constructive comment. We apologize for the misunderstanding arising due to our previous expression without presenting the meaning of vertical arrows. In Figure 2 (a and b), the vertical arrows point to the value of $(b/c)^*$ analytically derived, and we have clarified this point in the revised version. Beyond showing the effects of incomplete information on the fixation of cooperation in the pairwise social dilemmas, in these two panels, we also demonstrate a strong agreement between simulation results and theoretical calculations. And this further lays the foundation of the validity of our theory. Regarding the relaxation, in order to make the comparison with existing results, we perform analytical calculations under weak selection as well, i.e., $\delta \rightarrow 0$. However, in numerical simulations, we must specify the value of the selection intensity. So, what relaxes in the simulations is that we need to specify a selection intensity δ which is weak enough to verify our analytical predictions. Here in our simulations, we set $\delta = 0.01$.

- In Figure 3 only two results of global clustering are tested. As a result, in my opinion Figure 3 does not fully stress that there is a critical clustering coefficient to determine the impact of social information and that this critical value is the one derived analytically. One alternative to extend the analysis is to test the obtained critical b/c ratio for several realizations of the Watts-Strogatz small-

world networks with an increasing p (probability of rewire). This allows to interpolate between high clustering (0.5) and very low clustering coefficient (for $p=1$).

We thank Reviewer #2 for the insightful comment and providing a really inspiring idea. Following the great logic, we find that the lowest clustering coefficient (i.e., $p = 1$ as the Reviewer mentioned) is the clustering coefficient for random networks $C_0 = \langle d \rangle / (N - 1)$ ($\langle d \rangle$ is the average degree), which is above the critical threshold C^* estimated for regular networks. Considering that the heterogeneity in the node degree of small-world networks may influence the critical threshold of the clustering coefficient, after extensive investigations, we do not identify a Watts-Strogatz small-world network with a sufficiently low clustering coefficient which can present the same phenomenon as that in Figure 3b.

Nevertheless, inspired by the Reviewer's directions, we present a method to construct a series of regular networks with a relatively continuous change of the corresponding clustering coefficient. At the beginning, we construct a regular network with $C = 0$. Specifically, we consider a network with a fixed number of nodes $N = 50$. Then, following the node indices, we connect every node to its neighbours at distances 1, 3, 5, 7, 9, and 11. We designate the node preceding node i as $i - 1$ and the succeeding node as $i + 1$. Consequently, we seek to establish a numerical arrangement relationship such that the node preceding 0 is $N - 1$ (i.e., $i - 1 = N - 1$ if $i = 0$), and the node succeeding $N - 1$ is 0 (i.e., $i + 1 = 0$ if $i = N - 1$). After the initialization, we iterate over each node i and remove edge $(i, i + 3)$ and $(i + 1, i + 6)$, and then add edge $(i, i + 6)$ and $(i + 1, i + 3)$, where (i, j) represents the edge between node i and j . For every two nodes traversed, we save the structure of the network and assign a corresponding network identifier. We pick the first 11 networks with increasing clustering coefficients (Fig. R2a). Through simulations and the subsequent estimation of the corresponding values with linear fitting, we confirm that there exists a threshold on regular networks where the impact of social information reverses (Fig. R2b).

Motivated by Reviewer #2's excellent comment, we have now presented that there is a critical clustering coefficient to determine the impact of social information in the revised main text, and we have also added the Supplementary Note 4 and Supplementary Figures S7-S8 in the revised Supplementary Information to show this point.

Figure R2: The threshold of the clustering coefficient on regular networks. **a**, The clustering

coefficient \mathcal{C} increases with the number of networks generated. The dashed line represents the threshold of \mathcal{C} , i.e., \mathcal{C}^* , as shown in Eq. (7) in the main text. **b**, Taking the situation where individuals treat both kinds of information equally ($\theta = 1/(s + 1)$) as an example, we conduct simulations on networks with different clustering coefficients \mathcal{C} and find the corresponding r^* by solving the equations resulting from a linear regression of $\rho_C - \rho_D$ as a function of r . We draw r^* as a function of the clustering coefficient \mathcal{C} . The data points represent the values of r^* obtained from the linear regression of the simulation results, and the solid lines correspond to the theoretical values of r^* calculated from Eq. (6). The vertical dashed line represents the threshold of \mathcal{C} , i.e., \mathcal{C}^* , as shown in Eq. (7) in the main text.

*Minor points:

-line 47: nature section: natural selection

Thanks. We have corrected this in the revised manuscript.

-line 95: the authors can present here the range of r that guarantees that the game played is a social dilemma

We thank the reviewer for pointing this out. In the revised manuscript, we have presented the range of r that guarantees the game played is a social dilemma, namely, $1 < r < n$, where n is the number of players.

-line 133: derive intuition: to derive an intuition for?

We have changed it to “to provide an intuition for the impact of incomplete information on the evolution of cooperation”.

-line 170: depends on only the size N , and the degree, d of the network.

Thank you, and we have corrected these in the revised manuscript.

-line 296: minimize: disregard? ignore?

Thanks for pointing this out. We have corrected these in the revised manuscript and changed it into ‘ignore’.

-line 312: I wonder whether this can also be an example of “status quo bias”

We thank the reviewer for providing this insightful comment. After investigations, we find that “status quo bias” is indeed more suitable and we have used it following your suggestion in the revised manuscript.

-methods: In the beginning of the methods section the authors should recall the meaning of b and c , as well as the relevance of long random walks for the computational of an individual's payoff.

Thanks. We have done it in the revised manuscript following your meaningful comment.

-code availability: Regarding code availability the authors report the IDE that was used to code the analysis performed and do not detail the custom data that was created. It would be more relevant, to replicate the results, to report on the version of Julia that was used and, eventually, the computational resources required (such as time and memory) to replicate the findings.

We thank the reviewer for pointing this out. We have now added relevant information to replicate our results in the revised readme.txt file of our corresponding GitHub files. Please refer to <https://github.com/leizhougetbetter/IncompleteInfoImitation> for details

-Figure 3: some labels are impossible to read given the small font.

Good point. In the revised manuscript, we have enlarged both the font and labels in all figures.

Finally, we thank again Reviewer #2 for reviewing our manuscript. His/her insightful and constructive comments have considerably improved our manuscript.

Reviewers' Comments:

Reviewer #2:

Remarks to the Author:

The authors provided clear answers and revised the manuscript accordingly. I would like to thank them for that.

In my first review I listed concerns related to the methods' presentation, the pertinence of the numerical simulations and the generality of results regarding clustering coefficient. The authors managed to add content and clarifications that satisfy all my previous concerns. The new analysis on the critical values of clustering coefficient adds value to the manuscript. I believe this paper can be accepted.

Some final minor comments on the new content:

- The authors refer to "identical" networks; perhaps homogeneous networks, or networks with a delta degree distribution would be more appropriate (e.g., new Supplementary Figure S3)
- for Supplementary Figures S7-S8: I found a bit confusing using #networks as the independent variable in Figure R2; understanding the meaning of #networks might be challenging; a clearer measure could be "number of links rewired".

Reviewer #2 (Remarks to the Author):

The authors provided clear answers and revised the manuscript accordingly. I would like to thank them for that.

In my first review I listed concerns related to the methods' presentation, the pertinence of the numerical simulations and the generality of results regarding clustering coefficient. The authors managed to add content and clarifications that satisfy all my previous concerns. The new analysis on the critical values of clustering coefficient adds value to the manuscript. I believe this paper can be accepted.

We thank Reviewer #2 for his/her valuable suggestions, and for supporting the acceptance of our paper.

Some final minor comments on the new content:

- The authors refer to “identical” networks; perhaps homogeneous networks, or networks with a delta degree distribution would be more appropriate (e.g., new Supplementary Figure S3)

Thanks. We have changed “identical” to “homogeneous” in the main text and Supplementary information.

- for Supplementary Figures S7-S8: I found a bit confusing using #networks as the independent variable in Figure R2; understanding the meaning of #networks might be challenging; a clearer measure could be “number of links rewired” .

Thanks. In each step, there are two links to be rewired, and thus we have changed it to the “Pair of edges rewired”.